# APP: Adaptive Pose Pooling for 3D Human Pose Estimation from Videos

## ABSTRACT

Current advancements in 3D human pose estimation have attained notable success by converting 2D poses into their 3D counterparts. However, this approach is inherently influenced by the errors introduced by 2D pose detectors and overlooks the intrinsic spatial information embedded within RGB images. To address these challenges, we introduce a versatile module called Adaptive Pose Pooling (APP), compatible with many existing 2D-to-3D lifting models. The APP module includes three novel sub-modules: Pose-Aware Offsets Generation (PAOG), Pose-Aware Sampling (PAS), and Spatial Temporal Information Fusion (STIF). First, we extract latent features of the multi-frame lifting model. Then, a 2D pose detector is utilized to extract multi-level feature maps from the image. After that, PAOG generates offsets according to featuremaps. PAS uses offsets to sample featuremaps. Then, STIF can fuse PAS sampling features and latent features. This innovative design allows the APP module to simultaneously capture spatial and temporal information. We conduct comprehensive experiments on two widely used datasets: Human3.6M and MPI-INF-3DHP. Meanwhile, we employ various lifting models to demonstrate the efficacy of the APP module. Our results show that the proposed APP module consistently enhances the performance of lifting models, achieving state-of-the-art results. Significantly, our module achieves these performance boosts without necessitating alterations to the architecture of the lifting model. Code and checkpoints are available at: Anonymous Github.

## CCS CONCEPTS

• **Computing methodologies** → **Activity recognition and understanding**.

## KEYWORDS

3D Human Pose Estimation, Adaptive Pose Pooling, Feature Fusion

## 1 INTRODUCTION

3D human pose estimation (HPE) is a critical task in computer vision, and its goal is to estimate 3D human poses from images, videos, and 2D human poses. It attempts to estimate precise positions and orientations of 3D human keypoints. The applications of 3D HPE are pretty large, such as virtual/augmented reality [24, 29], motion analysis [2, 8, 43], human-computer interaction [1, 12], and autonomous driving [13] etc. Models for 3D HPE can be broadly

**Unpublished working draft. Not for distribution.**

Permission to make digital or hard copies of all or part of this work for personal or classroom use is granted without fee provided that copies are not made or distributed for profit or commercial advantage and that copies bear this notice and the full citation on the first page. Copyrights for components of this work owned by others than the author(s) must be honored. Abstracting with credit is permitted. To copy otherwise, or republish, to post on servers or to redistribute to lists, requires prior specific permission and/or a fee. Request permissions from permissions@acm.org.

*ACM MM, 2024, Melbourne, Australia*

© 2024 Copyright held by the owner/author(s). Publication rights licensed to ACM.
ACM ISBN 978-x-xxxx-xxxx-x/YY/MM
https://doi.org/10.1145/nnnnnnn.nnnnnnn

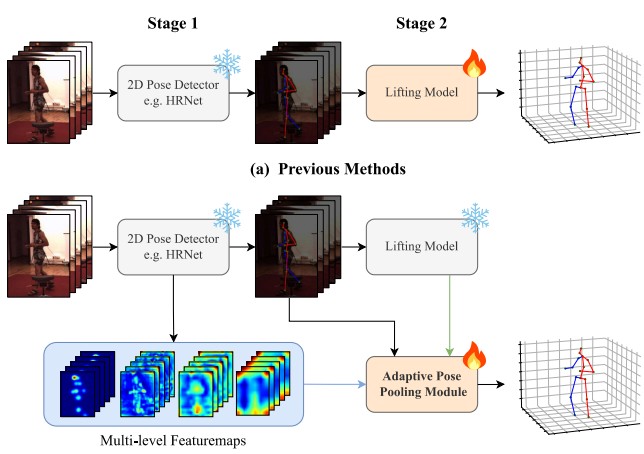

**Stage 1** **Stage 2**

2D Pose Detector e.g. HRNet — Lifting Model

**(a) Previous Methods**

2D Pose Detector e.g. HRNet — Lifting Model

Adaptive Pose Pooling Module

Multi-level Featuremaps

**(b) Our Method**

**Figure 1: Difference between our method and previous methods. (a) Previous methods freeze the 2D pose detector and train the lifting model. (b) Our proposed method utilizes the 2D pose features extracted by the lifting model and takes multi-level feature maps extracted by the 2D pose detector as inputs. Our method addresses the issue of underutilizing image features by leveraging multi-level feature maps extracted by a pretrained 2D pose detector. Furthermore, while single-frame models utilize image features, they lack temporal information. However, the APP module addresses this by extracting temporal and spatial information simultaneously.**

categorized into two fundamental classes based on the number of viewpoints employed: multi-view and single-view. Multi-view approaches leverage information from multiple camera perspectives, as explored in studies such as [6, 14, 17, 34, 46]. They mitigate depth ambiguity and reach better results than monocular 3D HPE methods. However, their practical applicability is hindered by the necessity of employing multiple cameras.

In contrast, single-view models offer a more applicable alternative and can be subdivided into single-stage and two-stage methodologies. Single-stage methods [3, 18, 25, 31, 40, 50] directly infer 3D human pose from input images or video frames without intermediate steps. While two-stage like early methods [26, 32] adopt a sequential approach. As shown in Figure 1(a), off-the-shelf 2D pose detectors [5, 30, 39] extract 2D human poses from input images or video frames. After that, they can obtain corresponding 3D human poses based on these 2D estimations, so this stage is often called the lifting stage. Models in the lifting stage can further be categorized into single-frame or multi-frame variants. Single-frame models [7, 47] infer 3D human poses from individual images, while multi-frame models process sequences of 2D human poses. The latter one contains two categories: many-to-one and many-to-many

methods. Many-to-one models estimating the 3D human pose of a central frame using multiple 2D human poses [21, 22, 48, 49], and many-to-many, also called seq2seq models inferring every frame of a 3D human pose from corresponding 2D human poses [4, 27, 33, 37, 45, 51]. They try to address different problems, ranging from spatio-temporal information exploration to enhancing model robustness.

In this paper, we focus on monocular 3D HPE. It uses only one camera to obtain images of humans. Therefore, it introduces ambiguity because of the lack of depth information, which is an ill-posed task. Besides, occlusion, including self-occlusion and occlusion by other objects and persons, needs to be drawn to researchers' attention. Hence, single-view models sometimes fail to get accurate and robust 3D human poses owing to depth ambiguity and occlusion. **Many monocular 3D HPE models suffer from them because the information of the input images fails to be fully used.** HEMlets Pose [50] and Context-Aware PoseFormer (CA-PF) [47] try to handle this. HEMlets Pose explicitly lets the model learn relative depth among different keypoints in 2D human poses. In comparison, CA-PF adopts Deformable Attention [52] to extract features of feature maps obtained by the 2D pose detector. However, these methods [47, 50] are **single-frame models that cannot capture temporal dynamics**. Recently, multi-frame lifting models have reached significant improvement. **Why not use both image and temporal features?** This work proposes a well-designed Adaptive Pose Pooling (APP) module. Serving as a plug-and-play component that can be integrated with many existing lifting models. The APP module facilitates extracting temporal and spatial information from multi-frame 2D human pose data and feature maps. Figure 1 illustrates the distinction between previous methods and our proposed method. Previous methods freeze the 2D pose detector and only take detected 2D poses as the model input to estimate corresponding 3D poses. Our method uses feature maps extracted by a 2D pose detector and takes multi-frame lifting models' calculated pose features. Therefore, the APP module inherently can capture spatio-temporal features. Do we really need to use all the features of the feature maps? Features near 2D poses should be more important than features at other positions. To enhance the model's flexibility, the model should adjust the sampling points during training. In the meantime, getting spatial and temporal features and interacting among features is also important for the model.

Our proposed APP module contains three novel submodules: Pose-Aware Offsets Generation (PAOG), Pose-Aware Sampling (PAS), and Spatial Temporal Information Fusion (STIF). To address the problem of insufficient utilization of spatial information in images, we propose PAOG and PAS. PAOG leverages 2D human poses as indices to learn offsets during training. PAS adaptively extracts features from feature maps generated by the 2D human pose detector. Though single-frame models use image features, they suffer from lacking temporal features. We proposed STIF to solve this problem by fusing image and temporal features. We conduct extensive experiments on two popular datasets: Human3.6M and MPI-INF-3DHP. Our proposed method achieves a new state-of-the-art performance, with 1.3mm (from 37.5mm to 36.2mm) and 1.1mm (from 30.6mm to 29.5mm) improvement in terms of Mean Per-Joint Position Error (MPJPE) and P-MPJPE (Procrustes-MPJPE) on Human3.6M. There are 4.6 (from 85.9 to 89.5) and 4mm (from 16.7mm to 12.7mm) improvements for Area Under Curve (AUC) and MPJPE on MPI-INF-3DHP.

Compared to current methods for monocular 3D HPE, our contributions are summarized as follows:

- We propose the APP module, enabling multi-frame lifting models to utilize information from the previous stage. This module is a plug-and-play solution compatible with most multi-frame lifting models and mitigates the problem of excessive reliance on 2D human pose outputs.
- We introduce Pose-Aware Offsets Generation (PAOG) and Pose-Aware Sampling (PAS). PAOG adjusts offsets relative to 2D human poses during training, allowing PAS to learn to adaptively capture spatial features from feature maps.
- We employ Spatial Temporal Information Fusion (STIF) to facilitate information interaction between the features sampled by PAS and the hidden feature of the lifting model. This enables the APP module to capture both temporal and spatial features simultaneously.
- We conduct experiments on two widely used datasets: Human3.6M and MPI-INF-3DHP. Our proposed approach achieves state-of-the-art performance on both datasets compared to previous methods. Importantly, it does not alter the lifting model's architecture.

## 2 RELATED WORK

Transformers[42] initially dominated the domain of natural language processing. Owing to its expressive performance, it has gained significant attention in computer vision as well [10]. Therefore, Transformer-based models have been increasingly applied across various domains, including 3D human pose estimation. Pose-Former [49] stands out as the pioneering model to integrate Transformers into 3D human pose estimation. Building upon this foundation, PoseFormerV2 [48] represents an advancement, introducing a frequency domain approach to enhance model robustness against fast motion changes. Furthermore, MHFormer [22] addresses the challenge of depth ambiguity by learning the representations of multiple hypothesis poses.

In the quest for improved efficiency and accuracy, novel architectural modifications have emerged. StridedTrans [21] utilizes fully connected layers in the Transformer encoder with stride convolutions. It reduces computational complexity and progressively shortens sequence lengths, thus enhancing pose estimation accuracy for the central frame. Meanwhile, MixSTE [45] focuses on modeling strong intra-frame correspondences between individual joints, thereby facilitating the learning of spatio-temporal correlations. P-STMO [37] adopts a strategy of occlusion masking to enable the model to learn more resilient pose representations. Additionally, it leverages a large volume of unlabeled data for self-supervised pre-training, thereby augmenting generalization capabilities. HoT [23] achieves this by implementing feature compression and restoration techniques, rendering it compatible with various lifting models while reducing computational overhead.

Advancements in attention mechanisms have also been pivotal. HDFormer [4], for instance, integrates self-attention and higher-order attention mechanisms to construct a hierarchical attention module aimed at mitigating complexities and handling heavily

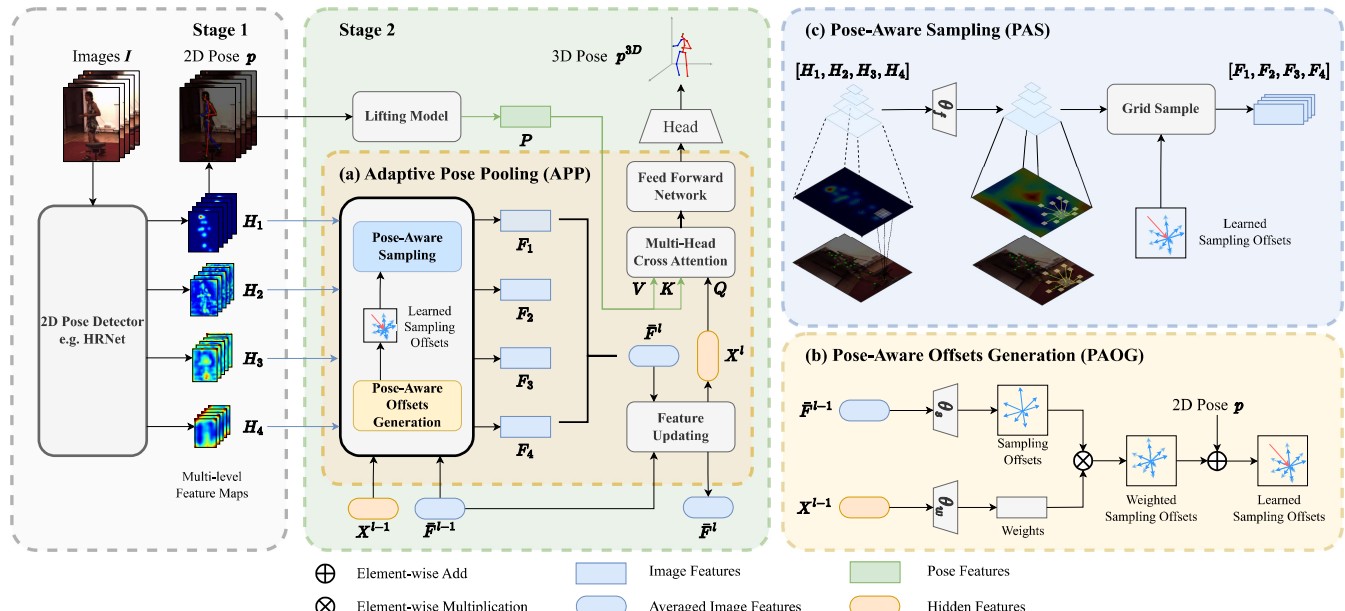

**Figure 2: Our proposed APP module. (a) APP module contains three novel sub-modules: Pose-Aware Offsets Generation (PAOG), Pose-Aware Sampling (PAS), and Spatial Temporal Information Fusion (STIF). STIF comprises Feature Updating and Multi-Head Cross Attention, which is not shown in this figure. Feature Updating is used to obtain $\bar{F}^l$ and $X^l$. MHCA fuses the hidden features $X^l$ with pose features $P$. (b) PAOG learns sampling offsets and utilizes these offsets to guide feature extraction. (c) PAS leverages learned offsets of PAOG and 2D human poses to sample multi-level feature maps.**

occluded scenarios. STCFormer [41] decomposes correlation learning into spatial and temporal dimensions, thereby reducing model complexity. Moreover, MotionBERT [51] emerges as a dual-stream spatio-temporal Transformer capable of concurrently capturing spatio-temporal information, showcasing state-of-the-art performance across multiple downstream tasks. Similarly, MotionAG-Former [27] introduces a hybrid architecture composed of multiple dual-stream modules, effectively capturing both spatial and temporal features of keypoints in tandem. KTPFormer[33] tries to overcome the weakness in existing transformer-based methods by proposing two plug-and-play attention modules, namely Kinematics Prior Attention (KPA) and Trajectory Prior Attention (TPA). These modules take advantage of the structure of the human body and motion trajectory information.

For the 2D-to-3D lifting stage, given a sequence of 2D human poses $p \in \mathbb{R}^{T \times J \times C_{in}}$, the goal is to estimate the corresponding 3D human poses $\hat{p} \in \mathbb{R}^{T \times J \times C_{out}}$, where $T$ is the sequence length of 2D pose frames, $J$ is the number of keypoints of the 2D human pose, $C_{in}$ and $C_{out}$ are the dimension of the 2D and 3D human pose keypoints, respectively.

These advancements collectively underscore the profound impact and ongoing evolution of Transformer-based approaches in advancing the field of 3D human pose estimation. Unlike our proposed method, which integrates image features into estimating 3D human poses, none of the existing transformer-based multi-frame methods take advantage of such information. This presents a unique opportunity for our approach to bridge this gap and potentially unlock new avenues for improving the accuracy and robustness of 3D human pose estimation methods.

## 3 APPROACH

### 3.1 Framework Overview

Figure 2 illustrates the overview of our proposed method. Given input images $I \in \mathbb{R}^{t \times 3 \times H \times W}$, where $t$ is the number of image frames and $H$, $W$ are the height and width of the image, note that $t \ll T$. We first use a 2D pose detector to extract the multi-level feature maps denoted by $\{H_i\}_{s=1}^{S}$, where $S$ is the number of scales. $H_1$ is a heatmap to get the 2D pose $p$. Then pose features $P \in \mathbb{R}^{T \times J \times d_P}$ is calculated in the multi-frame lifting models without regression head, where $d_P$ is the dimension of pose features. This process is formulated as:

$$\{H_i\}_{s=1}^{S} = Backbone(I),$$
$$P = MLM(p) \tag{1}$$

$MLM(\cdot)$ denotes multi-frame lifting models without regression head; the backbone is a 2D pose detector. In Figure 2, the values of averaged image features $\bar{F} \in \mathbb{R}^{T \times (k^2 \times J) \times d}$ and hidden features $X \in \mathbb{R}^{T \times J \times d}$ are updated by each layer of the APP module, where $k$ is the window size.

To get the initialized offsets $\Delta^0 \in \mathbb{R}^{T \times (k^2 \times J) \times C_{in}}$, we repeat the values by taking each point in the 2D human poses $p$ as a center and generating a $k \times k$ window around it. Then the $p$ is resized to $\mathbb{R}^{T \times (k^2 \times J) \times C_{in}}$. Now, the 2D poses $p$ data is a 2D vector, which means the $x$ and $y$ coordinates of the human keypoints.

We denote $\bar{F}^{l-1}$ and $X^{l-1}$ as the last layer outputs of the APP module. As for the first layer of APP, the offsets $\Delta^0$ are used to extract the initialized $\bar{F}^0$ and $X^0$:

$$
\begin{aligned}
\{H_s\}_{s=1}^{S} &= \theta_f^0(\{H_s\}_{s=1}^{S}), \\
\{F_s\}_{s=1}^{S} &= GridSample(\{H_s\}_{s=1}^{S}, \Delta^0), \\
\bar{F}^0 &= \frac{1}{S} \sum_{s=1}^{S} F_s, \\
X^0 &= AveragePool(\bar{F}^0)
\end{aligned}
\tag{2}
$$

Where $\theta_f$ is a group of 2D convolution operators to project multi-level feature maps into the same dimension $d$, the implementation of $GridSample$ is consistent with **nn.functional.grid_sample** in PyTorch, $AveragePool$ is to get the mean value of each $k \times k$ window.

Our proposed APP module consists of $L$ stacked layers. In detail, the calculation process of each layer APP module is defined as follows:

$$
\bar{F}^l, X^l = APP^l(\bar{F}^{l-1}, X^{l-1}, \boldsymbol{p}, \{H_i\}_{s=1}^{S}, \boldsymbol{P})
\tag{3}
$$

In the last layer of the APP module, a simple regression head consisting of two fully connected layers estimates the final 3D human pose given by $X^l$.

## 3.2 Adaptive Pose Pooling

### 3.2.1 Pose-Aware Offsets Generation.
Inspired by deformable convolution and deformable attention [9, 52], we propose the PAOG module as shown in Figure 2 (b). PAOG adaptively learns offset points during training. It takes the previous layer features $\bar{F}^{l-1}$ and $X^{l-1}$, and the 2D poses $P_p^{2D}$ as input to generate learned offsets $\Delta^l$, which is defined as follows:

$$
\begin{aligned}
\Delta_p^l &= PAOG^l(\bar{F}^{l-1}, X^{l-1}, \boldsymbol{p}) \\
&= \theta_s^l(\bar{F}^{l-1}) \otimes \theta_w^l(X^{l-1}) + \boldsymbol{p}
\end{aligned}
\tag{4}
$$

Where $\otimes$ denotes element-wise multiplication, $\theta_s^l$ and $\theta_w^l$ generate offset points based on the image center and their corresponding weights. Note that they are both fully connected layers.

### 3.2.2 Pose-Aware Sampling.
Using the offset points $\Delta_p^l$ generated by PAOG and the previous averaged image features $\bar{F}^{l-1}$, PAS also utilizes the multi-level feature maps $\{H\}_{s=1}^{S} \in \mathbb{R}^{t \times C_s \times H_s \times W_s}$ extracted from the 2D pose detector. We follow Deformable-DETR [52] before sampling features from the multi-level feature maps $F$. They are projected into the same dimension $d$ as in Eq. (2). The sampling process is then defined as:

$$
\begin{aligned}
\{H_s\}_{s=1}^{S} &= \theta_f^l(\{H_s\}_{s=1}^{S}), \\
\{F_s\}_{s=1}^{S} &= GridSample(\{H_s\}_{s=1}^{S}, \Delta^l), \\
\bar{F}^l &= \frac{1}{S} \sum_{s=1}^{S} F_s
\end{aligned}
\tag{5}
$$

### 3.2.3 Feature Updating.
At the $l$ layer of the APP module, we update the averaged image features $\bar{F}^l$ and hidden features $X^l$ as follows:

$$
\begin{aligned}
X^l &= \theta_k^l(\bar{F}^l), \\
\bar{F}^l &= \alpha \bar{F}^{l-1} + (1 - \alpha)\bar{F}^l
\end{aligned}
\tag{6}
$$

Where $\theta_k^l(\cdot)$ represents a convolution operation with a kernel size of $k^2$, whose size is the same as the previously mentioned $k \times k$ window, this operation allows the PAS to learn the weights of $k^2$ features within the window to calculate $X^l$ given by $\bar{F}^l$. $\alpha$ is a parameter controlling the magnitude of the update for $\bar{F}^l$.

### 3.2.4 Multi-Head Cross Attention.
MHCA enables the APP module to harness the temporal features extracted by the multi-frame lifting models while interacting with the spatial features extracted by PAS. We first combine the pose feature $\boldsymbol{P}$ extracted with positional embeddings $PE$, initialized to zero, and updated during training. We take $X^l$ as $\boldsymbol{Q}$, $\boldsymbol{P}$ as $\boldsymbol{K}$, and $\boldsymbol{V}$, then we perform as follows:

$$
\begin{aligned}
\boldsymbol{Q} &= X^l \boldsymbol{W_Q}, \boldsymbol{K} = \boldsymbol{P}\boldsymbol{W_K}, \boldsymbol{V} = \boldsymbol{P}\boldsymbol{W_V}, \\
X^l &= MHCA^l(\boldsymbol{Q}, \boldsymbol{K}, \boldsymbol{V}) \\
&= Concat(head_1, ..., head_h), \\
head_j &= Softmax\left(\frac{\boldsymbol{Q}_j \boldsymbol{K}_j}{\sqrt{d_P}}\right)\boldsymbol{V}_j
\end{aligned}
\tag{7}
$$

Where $\boldsymbol{W_Q} \in \mathbb{R}^{d \times d}$, $\boldsymbol{W_K} \in \mathbb{R}^{d_P \times d}$, and $\boldsymbol{W_V} \in \mathbb{R}^{d_P \times d}$, $j (j = 1, ..., h)$ represents the $j$-th attention head, and $h$ is the number of attention heads.

### 3.2.5 Feed Forward Network.
Like many monocular 3D HPE transformer-based methods, we apply an FFN at the end to finish computing the current layer. We use GeLU [15] as the activation function, as outlined in Eq. (8).

$$
\boldsymbol{H}^l = FFN^l = FC(GeLU(FC(X^l)))
\tag{8}
$$

### 3.2.6 Loss Function.
The loss function of our proposed APP module is the same as [27, 51]. In addition to the widely used MPJPE loss, we also incorporate the Mean Per-Joint Velocity Error (MPJVE) loss and the normalized MPJPE [36] loss. Let $\mathcal{L}_p$, $\mathcal{L}_v$, and $\mathcal{L}_s$ denote these three losses. $\mathcal{L}_p$ calculates the $L_2$ distance between estimated 3D human poses $\hat{\boldsymbol{p}}$ and ground truth $\boldsymbol{g}$, while $\mathcal{L}_v$ minimizes the difference between $\Delta\hat{\boldsymbol{p}}$ and $\Delta\boldsymbol{g}$. $\mathcal{L}_p$ and $\mathcal{L}_v$ are thus given by:

$$
\begin{aligned}
\mathcal{L}_p &= \sum_{t=1}^{T} \sum_{j=1}^{J} \|\hat{\boldsymbol{p}}_{tj} - \boldsymbol{g}_{tj}\|_2, \\
\mathcal{L}_v &= \sum_{t=2}^{T} \sum_{j=1}^{J} \|\Delta\hat{\boldsymbol{p}}_{tj} - \Delta\boldsymbol{g}_{tj}\|_2
\end{aligned}
\tag{9}
$$

Where $\Delta\hat{\boldsymbol{p}}_t = \hat{\boldsymbol{p}}_t - \hat{\boldsymbol{p}}_{t-1}$, $\Delta\boldsymbol{g}_t = \boldsymbol{g}_t - \boldsymbol{g}_{t-1}$. For $\mathcal{L}_s$, estimated 3D poses $\hat{\boldsymbol{p}}$ is scaled according to ground truth $\boldsymbol{g}$. Then, we calculate MPJPE loss between scaled $\hat{\boldsymbol{p}}$ and $\boldsymbol{g}$. The total loss is formulated as follows:

$$
\mathcal{L} = \gamma_p \mathcal{L}_p + \gamma_v \mathcal{L}_v + \gamma_s \mathcal{L}_s
\tag{10}
$$

where $\gamma_p$, $\gamma_v$, and $\gamma_s$ are weights for $\mathcal{L}_p$, $\mathcal{L}_v$, and $\mathcal{L}_s$, respectively.

**Table 1: Qualitative comparisons of 3D human pose estimation per action on Human3.6M. The best and second-best results are bolded and blue, respectively. *T*: Number of the input frames. Seq2Seq: Estimate 3D human pose for every frame.**

| MPJPE | Seq2Seq | T | Dir. | Disc | Eat | Greet | Phone | Photo | Pose | Purch. | Sit | SitD. | Smoke | Wait | WalkD. | Walk | WalkT. | Avg. |
|---|---|---|---|---|---|---|---|---|---|---|---|---|---|---|---|---|---|---|
| MHFormer [22] CVPR'22 | | 351 | 39.2 | 43.1 | 40.1 | 40.9 | 44.9 | 51.2 | 40.6 | 41.3 | 53.5 | 60.3 | 43.7 | 41.1 | 43.8 | 29.8 | 30.6 | 43.0 |
| StridedTrans [21] TMM'22 | | 351 | 40.3 | 43.3 | 40.2 | 42.3 | 45.6 | 52.3 | 41.8 | 40.5 | 55.9 | 60.6 | 44.2 | 43.0 | 44.2 | 30.0 | 30.2 | 43.7 |
| MixSTE [45] CVPR'22 | ✓ | 243 | 37.6 | 40.9 | 37.3 | 39.7 | 42.3 | 49.9 | 40.1 | 39.8 | 51.7 | 55.0 | 42.1 | 39.8 | 41.0 | 27.9 | 27.9 | 40.9 |
| MixSTE [45] CVPR'22 | ✓ | 243 | 36.7 | 39.0 | 36.5 | 39.4 | 40.2 | 44.9 | 39.8 | 36.9 | 47.9 | 54.8 | 39.6 | 37.8 | 39.3 | 29.7 | 30.6 | 39.8 |
| P-STMO [37] ECCV'22 | | 243 | 38.9 | 42.7 | 40.4 | 41.1 | 45.6 | 49.7 | 40.9 | 39.9 | 55.5 | 59.4 | 44.9 | 42.2 | 42.7 | 29.4 | 29.4 | 42.8 |
| CA-PF-HRNet-48 [47] NeurIPS'23 | | 1 | - | - | - | - | - | - | - | - | - | - | - | - | - | - | - | 39.8 |
| HDFormer [4] IJCAI'23 | ✓ | 96 | 38.1 | 43.1 | 39.3 | 39.4 | 44.3 | 49.1 | 41.3 | 40.8 | 53.1 | 62.1 | 43.3 | 41.8 | 43.1 | 31.0 | 29.7 | 42.6 |
| HDFormer [4] IJCAI'23 | ✓ | 96 | 34.7 | 41.7 | 36.0 | 38.4 | 41.1 | 45.3 | 39.6 | 37.4 | 49.0 | 63.1 | 39.8 | 38.9 | 40.2 | 29.3 | 29.1 | 40.3 |
| PoseFormerV2 [48] (f=27) CVPR'23 | | 243 | - | - | - | - | - | - | - | - | - | - | - | - | - | - | - | 45.2 |
| STCFormer [41] CVPR'23 | ✓ | 243 | 39.6 | 41.6 | 37.4 | 38.8 | 43.1 | 51.1 | 39.1 | 39.7 | 51.4 | 57.4 | 41.8 | 38.5 | 40.7 | 27.1 | 28.6 | 41.0 |
| STCFormer-L [41] CVPR'23 | ✓ | 243 | 38.4 | 41.2 | 36.8 | 38.0 | 42.7 | 50.5 | 38.7 | 38.2 | 52.5 | 56.8 | 41.8 | 38.4 | 40.2 | 26.2 | 27.7 | 40.5 |
| UPS [11] CVPR'23 | | 243 | 37.5 | 39.2 | 36.9 | 40.6 | 39.3 | 46.8 | 39.0 | 41.7 | 50.6 | 63.5 | 40.4 | 37.8 | 44.2 | 26.7 | 29.1 | 40.8 |
| D3DP [38] (H=20,K=10, J-Agg) ICCV'23 | ✓ | 243 | 37.3 | 39.5 | 35.6 | 37.8 | 41.3 | 48.2 | 39.1 | 37.6 | 49.9 | 52.8 | 41.2 | 39.2 | 39.4 | 27.2 | 27.1 | 39.5 |
| GLA-GCN [44] ICCV'23 | | 243 | 41.3 | 44.3 | 40.8 | 41.8 | 45.9 | 54.1 | 42.1 | 41.5 | 57.8 | 62.9 | 45.0 | 42.8 | 45.9 | 29.4 | 29.9 | 44.4 |
| MotionBERT [51] (scratch) ICCV'23 | ✓ | 243 | 36.3 | 38.7 | 38.6 | 33.6 | 42.1 | 50.1 | 36.2 | 35.7 | 50.1 | 56.6 | 41.3 | 37.4 | 37.7 | 25.6 | 26.5 | 39.2 |
| MotionBERT [51] (finetune) ICCV'23 | ✓ | 243 | 36.1 | 37.5 | 35.8 | 32.1 | 40.3 | 46.3 | 36.1 | 35.3 | 46.9 | 53.9 | 39.5 | 36.3 | 35.8 | 25.1 | 25.3 | 37.5 |
| HoT [23] CVPR'24 | | 243 | - | - | - | - | - | - | - | - | - | - | - | - | - | - | - | 39.0 |
| KTPFormer [33] CVPR'24 | ✓ | 243 | 37.3 | 39.2 | 35.9 | 37.6 | 42.5 | 48.2 | 38.6 | 39.0 | 51.4 | 55.9 | 41.6 | 39.0 | 40.0 | 27.0 | 27.4 | 40.1 |
| MotionAGFormer-B [27] WACV'24 | ✓ | 243 | 36.4 | 38.4 | 36.8 | 32.9 | 40.9 | 48.5 | 36.6 | 34.6 | 51.7 | 52.8 | 41.0 | 36.4 | 36.5 | 26.7 | 27.0 | 38.4 |
| **APP w. MotionBERT (finetune) (Ours)** | ✓ | 243 | 34.6 | 36.7 | 37.4 | 31.5 | 38.9 | 45.7 | 35.9 | 33.3 | 48.4 | 53.4 | 39.1 | 36.3 | 34.4 | 24.2 | 24.8 | **37.0** |
| **APP w. MotionAGFormer-B (Ours)** | ✓ | 243 | 33.7 | 37.2 | 35.0 | 32.2 | 37.2 | 40.7 | 34.6 | 34.3 | 45.1 | 51.4 | 37.0 | 34.6 | 36.8 | 27.0 | 26.8 | **36.2** |
| **P-MPJPE** | | T | Dir. | Disc | Eat | Greet | Phone | Photo | Pose | Purch. | Sit | SitD. | Smoke | Wait | WalkD. | Walk | WalkT. | Avg. |
| MHFormer [22] CVPR'22 | | 351 | 31.5 | 34.9 | 32.8 | 33.6 | 35.3 | 39.6 | 32.0 | 32.2 | 43.5 | 48.7 | 36.4 | 32.6 | 34.3 | 23.9 | 25.1 | 34.4 |
| StridedTrans [21] TMM'22 | | 351 | - | - | - | - | - | - | - | - | - | - | - | - | - | - | - | - |
| MixSTE [45] CVPR'22 | ✓ | 243 | 30.8 | 33.1 | 30.3 | 31.8 | 33.1 | 39.1 | 31.1 | 30.5 | 42.5 | 44.5 | 34.0 | 30.8 | 32.7 | 22.1 | 22.9 | 32.6 |
| MixSTE [45] CVPR'22 | ✓ | 243 | 28.0 | 30.9 | 28.6 | 30.7 | 30.4 | 34.6 | 28.6 | 28.1 | 37.1 | 47.3 | 30.5 | 29.7 | 30.5 | 21.6 | 20.0 | 30.6 |
| P-STMO [37] ECCV'22 | | 243 | 31.3 | 35.2 | 32.9 | 33.9 | 35.4 | 39.3 | 32.5 | 31.5 | 44.6 | 48.2 | 36.3 | 32.9 | 34.4 | 23.8 | 23.9 | 34.4 |
| CA-PF-HRNet-48 [47] NeurIPS'23 | | 1 | - | - | - | - | - | - | - | - | - | - | - | - | - | - | - | 32.7 |
| HDFormer [4] IJCAI'23 | ✓ | 96 | 29.6 | 33.8 | 31.7 | 31.3 | 33.7 | 37.7 | 30.6 | 31.0 | 41.4 | 47.6 | 35.0 | 30.9 | 33.7 | 25.3 | 23.6 | 33.1 |
| HDFormer [4] IJCAI'23 | ✓ | 96 | 27.9 | 32.8 | 29.7 | 30.6 | 32.5 | 35.0 | 28.9 | 29.2 | 38.3 | 50.0 | 32.9 | 30.1 | 31.8 | 23.6 | 22.8 | 31.7 |
| PoseFormerV2 [48] (f=27) CVPR'23 | | 243 | - | - | - | - | - | - | - | - | - | - | - | - | - | - | - | - |
| STCFormer [41] CVPR'23 | ✓ | 243 | 29.5 | 33.2 | 30.6 | 31.0 | 33.0 | 38.0 | 30.4 | 29.4 | 41.8 | 45.2 | 33.6 | 29.5 | 31.6 | 21.3 | 22.6 | 32.0 |
| STCFormer-L [41] CVPR'23 | ✓ | 243 | 29.3 | 33.0 | 30.7 | 30.6 | 32.7 | 38.2 | 29.7 | 28.8 | 42.2 | 45.0 | 33.3 | 29.4 | 31.5 | 20.9 | 22.3 | 31.8 |
| UPS [11] CVPR'23 | | 243 | 30.3 | 32.2 | 30.8 | 33.1 | 31.1 | 35.2 | 30.3 | 32.1 | 39.4 | 49.6 | 32.9 | 29.2 | 33.9 | 21.6 | 24.5 | 32.5 |
| D3DP [38] (H=20,K=10, J-Agg) ICCV'23 | ✓ | 243 | 30.6 | 32.4 | 29.2 | 30.9 | 31.9 | 37.4 | 30.2 | 29.3 | 40.4 | 43.2 | 33.2 | 30.4 | 31.3 | 21.5 | 22.3 | 31.6 |
| GLA-GCN [44] ICCV'23 | | 243 | 32.4 | 35.3 | 32.6 | 34.2 | 35.0 | 42.1 | 32.1 | 31.9 | 45.5 | 49.5 | 36.1 | 32.4 | 35.6 | 23.5 | 24.7 | 34.8 |
| MotionBERT [51] (scratch) ICCV'23‡ | ✓ | 243 | 30.8 | 32.8 | 32.4 | 28.7 | 34.3 | 38.9 | 30.1 | 30.0 | 42.5 | 49.7 | 36.0 | 30.8 | 31.7 | 22.0 | 23.0 | 32.9 |
| MotionBERT [51] (finetune) ICCV'23 | ✓ | 243 | - | - | - | - | - | - | - | - | - | - | - | - | - | - | - | - |
| HoT [23] CVPR'24 | | 243 | - | - | - | - | - | - | - | - | - | - | - | - | - | - | - | - |
| KTPFormer [33] CVPR'24 | ✓ | 243 | 30.1 | 32.3 | 29.6 | 30.8 | 32.3 | 37.3 | 30.0 | 30.2 | 41.0 | 45.3 | 33.6 | 29.9 | 31.4 | 21.5 | 22.6 | 31.9 |
| MotionAGFormer-B [27] WACV'24 | ✓ | 243 | 30.6 | 32.6 | 32.2 | 28.2 | 33.8 | 38.6 | 30.5 | 29.9 | 43.3 | 47.0 | 35.2 | 29.8 | 31.4 | 22.7 | 23.5 | 32.6 |
| **APP w. MotionBERT (finetune) (Ours)** | ✓ | 243 | 30.0 | 31.3 | 31.9 | 26.9 | 32.8 | 37.0 | 29.8 | 29.2 | 40.7 | 47.9 | 34.1 | 30.0 | 30.0 | 21.0 | 21.7 | 31.6 |
| **APP w. MotionAGFormer-B (Ours)** | ✓ | 243 | 28.2 | 30.2 | 28.5 | 26.5 | 30.1 | 33.2 | 27.2 | 28.1 | 36.4 | 42.0 | 30.8 | 27.7 | 29.7 | 21.4 | 21.9 | **29.5** |

# 4 EXPERIMENTS

## 4.1 Datasets and Evaluation Metrics

*4.1.1 Human3.6M.* Human3.6M [16] is a widely used large-scale 3D human pose estimation dataset. The dataset consists of images captured by four high-resolution cameras placed at the corners of a room to capture the human body from multiple angles. It includes 11 subjects performing 17 action scenarios (e.g., talking on the phone, taking photos), making it a benchmark dataset for 3D human pose estimation. We follow previous studies [27, 51] by using subjects 1, 5, 6, 7, and 8 for model training and subjects 9 and 11 for testing.

*4.1.2 MPI-INF-3DHP.* MPI-INF-3DHP [28] dataset comprises indoor and outdoor scenes. It contains over 1.3 million frames captured from 14 camera viewpoints, recording eight actions performed by eight subjects. The test set includes seven action classes and three different scenarios: green screen, non-green screen, and outdoor scenes.

*4.1.3 Evaluation Metrics.* For the Human3.6M dataset, we report the MPJPE and P-MPJPE. Before computing the error, the former aligns the estimated 3D pose with the ground truth root node (hip). At the same time, the latter requires rigid alignment based on the ground truth, including translation and rotation, before error calculation. For the MPI-INF-3DHP dataset, consistent with previous works [4, 27, 38, 47, 48], we report MPJPE, Percentage of Correct Keypoints (PCK) with a threshold of 150mm, and Area Under Curve (AUC).

## 4.2 Implementation Details

We adopt Transformer [42], MotionAGFormer-B [27], MixSTE [45], MotionBERT [51], and HoT [23] as the foundation models. These models' dimensions $d_P$ are 128, 256, 512, and 128, respectively. For a fair comparison, we use pretrained lifting models provided by the authors.

YOLOv3 [35] is used to extract bounding boxes of humans in the images, and then we crop the images to $192 \times 256$ according to the bounding boxes. However, in [47], the 3D ground truth human pose is directly used to extract bounding boxes, which is not feasible in real-world scenarios.

**Table 2: Qualitative comparisons of 3D human pose estimation per action using 2D Ground Truth (GT) human poses on Human3.6M. The best and second-best results are bolded and blue, respectively. $T$: Number of the input frames. Seq2Seq: Estimate 3D human pose for every frame.**

| MPJPE (GT) | Seq2Seq | T | Dir. | Disc | Eat | Greet | Phone | Photo | Pose | Purch. | Sit | SitD. | Smoke | Wait | WalkD. | Walk | WalkT. | Avg. |
|---|---|---|---|---|---|---|---|---|---|---|---|---|---|---|---|---|---|---|
| MHFormer [22] CVPR'22 | | 351 | 27.7 | 32.1 | 29.1 | 28.9 | 30.0 | 33.9 | 33.0 | 31.2 | 37.0 | 39.3 | 30.0 | 31.0 | 29.4 | 22.2 | 23.0 | 30.5 |
| StridedTrans [21] TMM'22 | | 351 | 27.1 | 29.4 | 26.5 | 27.1 | 28.6 | 33.0 | 30.7 | 26.8 | 38.2 | 34.7 | 29.1 | 29.8 | 26.8 | 19.1 | 19.8 | 28.5 |
| MixSTE [45] CVPR'22 | ✓ | 81 | 25.6 | 27.8 | 24.5 | 25.7 | 24.9 | 29.9 | 28.6 | 27.4 | 29.9 | 29.0 | 26.1 | 25.0 | 25.2 | 18.7 | 19.9 | 25.9 |
| MixSTE [45] CVPR'22 | ✓ | 243 | 21.6 | 22.0 | 20.4 | 21.0 | 20.8 | 24.3 | 24.7 | 21.9 | 26.9 | 24.9 | 21.2 | 21.5 | 20.8 | 14.7 | 15.6 | 21.6 |
| P-STMO [37] ECCV'22 | | 243 | 28.5 | 30.1 | 28.6 | 27.9 | 29.8 | 33.2 | 31.3 | 27.8 | 36.0 | 37.4 | 29.7 | 29.5 | 28.1 | 21.0 | 21.0 | 29.3 |
| D3DP [38] (H=20,K=10, J-Agg) ICCV'23 | ✓ | 243 | 19.9 | 19.4 | 19.4 | 19.0 | 19.8 | 22.0 | 21.4 | 19.1 | 24.8 | 23.2 | 19.6 | 18.7 | 18.6 | 14.0 | 14.5 | 19.6 |
| GLA-GCN [44] ICCV'23 | | 243 | 20.1 | 21.2 | 20.0 | 19.6 | 21.5 | 26.7 | 23.3 | 19.8 | 27.0 | 29.4 | 20.8 | 20.1 | 19.2 | 12.8 | 13.8 | 21.0 |
| STCFormer [41] CVPR'23 | ✓ | 81 | 25.9 | 25.9 | 22.7 | 24.0 | 24.6 | 27.5 | 27.6 | 23.1 | 30.1 | 31.5 | 25.1 | 24.7 | 23.8 | 18.4 | 19.6 | 25.0 |
| STCFormer-L [41] CVPR'23 | ✓ | 243 | 20.8 | 21.8 | 20.0 | 20.6 | 23.4 | 25.0 | 23.6 | 19.3 | 27.8 | 26.1 | 21.6 | 20.6 | 19.5 | 14.3 | 15.1 | 21.3 |
| HDFormer [4] IJCAI'23 | ✓ | 96 | - | - | - | - | - | - | - | - | - | - | - | - | - | - | - | 21.6 |
| MotionBERT [51] (scratch) ICCV'23 | ✓ | 243 | 16.7 | 19.9 | 17.1 | 16.5 | 17.4 | 18.8 | 19.3 | 20.5 | 24.0 | 22.1 | 18.6 | 16.8 | 16.7 | 10.8 | 11.5 | 17.8 |
| MotionBERT [51] (finetune) ICCV'23 | ✓ | 243 | 15.9 | 17.3 | 16.9 | 14.6 | 16.8 | 18.6 | 18.6 | 18.4 | 22.0 | 21.8 | 17.3 | 16.9 | 16.1 | 10.5 | 11.4 | 16.9 |
| KTPFormer [33] CVPR'24 | ✓ | 243 | 18.8 | 17.4 | 18.1 | 17.7 | 18.3 | 20.6 | 19.6 | 17.7 | 23.3 | 22.0 | 18.7 | 17.0 | 16.8 | 12.4 | 13.5 | 18.1 |
| MotionAGFormer-B [27] WACV'24 | ✓ | 243 | - | - | - | - | - | - | - | - | - | - | - | - | - | - | - | 19.4 |
| **APP w. MotionAGFormer-B (Ours)** | ✓ | 243 | 18.2 | 20.6 | 18.4 | 17.9 | 19.5 | 21.3 | 20.7 | 20.6 | 25.2 | 25.7 | 19.3 | 18.2 | 17.4 | 11.3 | 12.1 | 19.1 |

The lengths of 2D human pose sequences $T$ on Human3.6M and MPI-INF-3DHP are set to 243 and 81, respectively, and images are uniformly sampled into sequence lengths $t = \frac{T}{9}$ for a trade-off between performance and efficiency. Additionally, $t$ must be divisible by $T$. We extract feature maps from images using HRNet-w32 [39] with four resolutions. It is chosen as the pose detector, $C_s = 32^{s+1} \in \{32, 64, 128, 256\}$, $H_s = \frac{H}{4^{s+1}} \in \{\frac{H}{4}, \frac{H}{8}, \frac{H}{16}, \frac{H}{32}\}$, $W_s = \frac{W}{4^{s+1}} \in \{\frac{W}{4}, \frac{W}{8}, \frac{W}{16}, \frac{W}{32}\}$.

The parameters of our proposed APP module are mainly determined by the number of layers $L$, the number of attention heads $h$, the kernel size $k$, the dimension $d$ of the APP module, and the dimension of hidden feature $F_p$ used in the selected lifting model. Also, we employ DropPath [20] with the probability $p$. For Human3.6M and MPI-INF-3DHP, $L$, $h$, $k$, $\alpha$, $p$, and $d$ are set to 6, 8, 3, 0.9, 0.2, and 128, respectively. We train the APP module using the Adam optimizer [19]. 2D ground truth human pose is sent to the model for MPI-INF-3DHP. In Eq. (9), the weights of each part of the loss are set to 1.0, 20.0, and 0.5, respectively. All experiments are conducted on one NVIDIA RTX 3090 GPU.

## 4.3 Qualitative Comparison of Human3.6M

We conducted comparative evaluations with recent state-of-the-art (SOTA) models on the Human3.6M dataset. The results, summarized in Table 1, underscore the superiority of our proposed APP module when integrated with MotionAGFormer-B, attaining SOTA performance. Specifically, in comparison with lifting models MixSTE [45] and HDFormer [4], which also utilize HRNet-detected 2D human pose as input, our approach achieves notable improvements of 3.6mm (from 39.8mm to 36.2mm) and 4.1mm (from 40.3mm to 36.2mm) in MPJPE, and 1.1mm (from 30.6mm to 29.5mm) and 2.2mm (from 31.7mm to 29.5mm) in terms of P-MPJPE, respectively. What's more, our proposed method surpasses CA-PF-HRNet-48 [47] 3.6mm (from 39.8mm to 36.2mm) in MPJPE, and 3.2mm (from 32.7mm to 29.5mm).

Although our method exhibits a slightly higher MPJPE of 1.3mm compared to MotionBERT [51] when ground truth 2D human pose is employed as input, it surpasses MotionAGFormer-B by 0.3mm.

These findings collectively attest to the efficacy and competitiveness of our proposed approach in advancing the state-of-the-art in monocular 3D human pose estimation.

**Table 3: Qualitative comparisons of 3D human pose estimation per action on MPI-INF-3DHP. We report AUC, MPJPE, and PCK. The best and second-best results are bolded and blue, respectively. $T$: Number of the input frames.**

| Method | T | AUC↑ | MPJPE↓ | PCK↑ |
|---|---|---|---|---|
| D3DP [38] (H=20,K=10, J-Agg) ICCV'23 | 243 | 78.2 | 29.7 | 97.7 |
| HDFormer [4] IJCAI'23 | 32 | 64.0 | 51.5 | 96.8 |
| HDFormer [4] IJCAI'23 | 96 | 72.9 | 37.2 | 98.7 |
| PoseFormerV2 [48] CVPR'23 | 81 | 78.8 | 27.8 | 97.9 |
| CA-PF-HRNet-32 [47] NeurIPS'23 | 1 | 75.4 | 32.7 | 98.0 |
| CA-PF-HRNet-48 [47] NeurIPS'23 | 1 | 76.3 | 31.4 | 98.2 |
| MotionAGFormer-B [27] WACV'24 | 81 | 84.2 | 18.2 | 98.3 |
| KTPFormer [33] CVPR'24 | 27 | 84.4 | 19.2 | 98.9 |
| KTPFormer [33] CVPR'24 | 81 | 85.9 | 16.7 | 98.9 |
| **APP w. MotionAGFormer-B (Ours)** | 81 | 89.5 | 12.7 | 98.9 |

## 4.4 Qualitative Comparison of MPI-INF-3DHP

In evaluating the MPI-INF-3DHP dataset, we aligned the sequence length $T$ to 81 frames to maintain consistency with the experimental setup of MotionAGFormer-B [27]. As detailed in Table 2, our method demonstrates incremental enhancements across various evaluation metrics. Specifically, we observe marginal improvements in the AUC and MPJPE upon KTPFormer [33], with gains of 3.6 (from 85.9 to 89.5) and 4mm (from 16.7mm to 12.7mm), respectively. Moreover, the PCK of our method is the same as KTPFormer [33] because the threshold of PCK is set to 150mm. When comparing our method with the baseline model MotionAGFormer-B[27], there are huge improvements for AUC (from 84.2 to 89.5) and MPJPE (from 18.2mm to 12.7mm). These findings underscore the efficacy and robustness of our proposed method in enhancing monocular 3D human pose estimation performance on the MPI-INF-3DHP dataset.

## 4.5 Ablation Study

We conduct extensive experiments to substantiate the effectiveness of our proposed APP module. As shown in Table 4, our experimental results manifest a consistent enhancement in the performance of the lifting models across diverse configurations of 2D pose detectors and multiple-frame lifting models, affirming the efficacy of the APP module. Transformer [42] is a simple spatial-temporal vanilla transformer with 12 layers.

**Table 4: In comparison with baselines on Human3.6M, we choose different lifting models and 2D poses.**

| Method | $T$ | 2D Pose | Param | MACs | MACs/frame | MPJPE↓ | P-MPJPE↓ |
|---|---|---|---|---|---|---|---|
| Transformer [42] NeurIPS'17 | 243 | SH | 9.62M | 46.16G | 189.95G | 39.2 | 32.8 |
| **APP w. Transformer (Ours)** | 243 | SH | 13.20M | 70.71G | 290.97G | 39.0 | 32.7 |
| MixSTE [45] CVPR'22 | 243 | CPN | 33.78M | 147.60G | 607.38G | 40.9 | 32.6 |
| **APP w. MixSTE (Ours)** | 243 | CPN | 39.54M | 174.85G | 719.54G | 39.9 | 32.1 |
| MotionBERT [51] (finetune) ICCV'23 | 243 | SH | 42.47M | 185.60G | 763.77G | 37.5 | - |
| **APP w. MotionBERT (finetune) (Ours)** | 243 | SH | 48.23M | 211.77G | 871.48G | 37.0 | 31.6 |
| MotionAGFormer-B [27] WACV'24 | 243 | CPN | 11.72M | 64.78G | 266.60G | 39.2 | 31.5 |
| **APP w. MotionAGFormer-B (Ours)** | 243 | CPN | 15.30M | 90.14G | 370.97G | 38.9 | 31.3 |
| MotionAGFormer-B [27] WACV'24 | 243 | SH | 11.72M | 64.78G | 266.60G | 38.4 | 32.6 |
| **APP w. MotionAGFormer-B (Ours)** | 243 | SH | 15.30M | 90.14G | 370.97G | 38.4 | 32.3 |
| MotionAGFormer-B [27] WACV'24 | 243 | HRNet | 11.72M | 64.78G | 266.60G | 36.6 | 29.5 |
| **APP w. MotionAGFormer-B (Ours)** | 243 | HRNet | 15.30M | 90.14G | 370.97G | 36.2 | 29.5 |
| MotionAGFormer-B [27] WACV'24 | 243 | GT | 11.72M | 64.78G | 266.60G | 19.4 | - |
| **APP w. MotionAGFormer (Ours)** | 243 | GT | 15.30M | 90.14G | 370.97G | 19.1 | 18.4 |
| HoT [23] CVPR'24 | 243 | SH | 16.35M | 33.48G | 137.79G | 39.8 | 33.5 |
| **APP w. HoT (Ours)** | 243 | SH | 20.66M | 50.80G | 209.06G | 39.7 | 33.4 |

Notably, the APP module contributes to performance improvements while maintaining small parameters, accounting for approximately 37% (from 9.62M to 13.2M), 17% (from 33.78M to 39.54M), 14% (from 42.47M to 48.23M), 30% (from 11.72M to 15.3M), 26% (from 16.35M to 20.66M) parameters of Transformer [42], MixSTE [45], MotionBERT [51], MotionAGFormer-B [27], HoT [23], respectively. When it comes to MACs, the gains are 34% (from 46.16G to 70.71G), 16% (from 147.6G to 174.85G), 12% (from 185.6G to 211.77G), 28% (from 64.78G to 90.14G), 34% (from 33.48G to 50.8G).

All ablation studies are conducted on the Human3.6M dataset, with MotionAGFormer-B [27] chosen as the baseline model and 2D human body pose detected using HRNet [39]. Initially, we conducted experiments on parameter selections for the APP module. The results of Table 5 presents indicate that our proposed method achieves the best performance when $L$, $k$, $\alpha$, $p$, and $d$ are selected as 6, 3, 0.9, 0.2, and 128, respectively.

**Table 5: Ablation study of the parameters on Human3.6M. $L$: The number of layers of our proposed APP module. $p$: Probability of the DropPath. $d$: The Number of the dimensions. We report MPJPE and P-MPJPE. The best and second-best results are bolded and blue, respectively.**

| $L$ | $k$ | $\alpha$ | $p$ | $d$ | Params | MPJPE↓ | P-MPJPE↓ |
|---|---|---|---|---|---|---|---|
| 6 | 3 | 0.9 | 0.2 | 128 | 3.58M | **36.2** | **29.5** |
| 4 | 3 | 0.9 | 0.2 | 256 | 7.50M | 36.5 | 29.7 |
| 4 | 3 | 0.9 | 0.4 | 128 | 2.76M | 36.3 | 29.6 |
| 6 | 5 | 0.9 | 0.2 | 128 | 5.15M | 36.5 | 29.7 |
| 6 | 3 | 0.6 | 0.2 | 128 | 3.58M | 36.6 | 30.0 |
| 6 | 3 | 0.9 | 0 | 128 | 3.58M | 36.5 | 29.9 |

Then, we conducted ablation experiments on the submodules of our proposed APP module. In Table 6, the first row presents partial components of three submodules: MHCA, PAOG, and PAS. The table shows that MHCA has the most significant impact on the APP model's performance. However, the other three components also enhance the model's performance to varying degrees.

**Table 6: Ablation study of the components on Human3.6M. MHCA: Use MHCA. Weights: Apply weights to the learned offsets. PoseAware: Choose the detected 2D pose as the pivot; otherwise, the center of the image will be selected. Conv: Convolution is performed on the sampled features; otherwise, the mean operation is carried out directly. We report MPJPE and P-MPJPE. The best and second-best results are bolded and blue, respectively.**

| MHCA | Weights | PoseAware | Conv | Params | MPJPE↓ | P-MPJPE↓ |
|---|---|---|---|---|---|---|
| ✓ | | | | 2.69M | 36.6 | 29.8 |
| | ✓ | | | 2.30M | 62.0 | 43.1 |
| | | ✓ | | 2.30M | 68.0 | 44.7 |
| | | | ✓ | 3.18M | 67.2 | 45.7 |
| ✓ | ✓ | | | 2.69M | **36.2** | **29.6** |
| ✓ | | ✓ | | 2.69M | 36.7 | 29.9 |
| ✓ | | | ✓ | 3.58M | 36.4 | **29.6** |
| | ✓ | ✓ | | 2.30M | 63.6 | 43.2 |
| | ✓ | | ✓ | 3.18M | 59.1 | 40.0 |
| | | ✓ | ✓ | 3.18M | 71.0 | 45.1 |
| ✓ | ✓ | ✓ | | 2.69M | 36.5 | 30.0 |
| ✓ | ✓ | | ✓ | 3.58M | **36.3** | 29.9 |
| ✓ | | ✓ | ✓ | 3.58M | 36.4 | **29.6** |
| | ✓ | ✓ | ✓ | 3.18M | 57.0 | 39.1 |
| ✓ | ✓ | ✓ | ✓ | 3.58M | **36.2** | **29.5** |

Finally, we conducted ablaton experiments on image frames $t$ setting. Although the MACs (24.83G) of the APP module are not affected by the number of frames, yet image frames $t$ need more computation in training stage because we need multi-level featuremaps extracted by 2D pose detector. Furthermore, when $t \geq 81$, the training time becomes very long. Therefore, $t = 27 = \frac{T}{9}$ is tradeoff between accuracy and efficiency. In inference stage, this problem no longer appears, owing to that 2D pose detector is used to estimate 2D pose for every frame.

**Table 7: Ablation study of the image frames $t$ on Human3.6M. We report MPJPE and P-MPJPE. The best and second-best results are bolded and blue, respectively.**

| $t$ | Params | MPJPE↓ | P-MPJPE↓ |
|---|---|---|---|
| 1 | 3.58M | 36.5 | 30.0 |
| 3 | 3.58M | 36.5 | **29.8** |
| 9 | 3.58M | **36.3** | 29.6 |
| 27 | 3.58M | **36.2** | 29.6 |

## 4.6 Visualization

To validate the generalization ability of the model, we selected the MotionBERT [51] (finetune version) and MotionAGFormer-B [27] as foundation models, which were trained on 2D poses extracted by SH [30] on Human3.6M dataset, for comparison with our proposed APP module. We chose two video segments from the internet to

visually analyze the APP module. For the visual analysis in Figures 3 and 4, MotionAGFormer-B was selected as the foundation model.

*4.6.1 Analysis of MHCA Module.* We visualized the attention maps of the MHCA submodule at each layer of our proposed APP module. The values of each attention map were normalized to $[0, 1]$. Each attention map extracted at every layer is a $17 \times 17$ matrix. Figure 3 shows that although each layer has different focuses, the APP module can capture the relationships between key points.

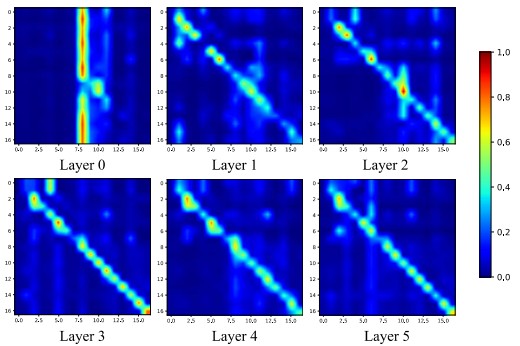

**Figure 3: Visualization of the attention maps of each layer of MHCA in our proposed APP module.**

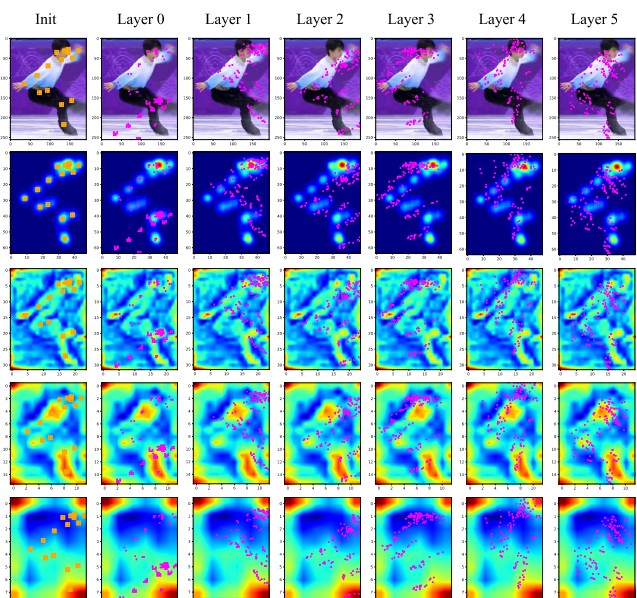

**Figure 4: Visualization of the initialized/learned sampling points learned by each layer, where the initialized/learned sampling points are marked in orange and magenta, respectively. Each sampling point is represented by a cross (x).**

*4.6.2 Analysis of PAOG Module.* Figure 4 illustrates the sampling points learned by our proposed PAOG submodule. The first row in the figure is appended for observation, while the second to fifth

rows represent feature maps extracted at different resolutions. Each layer of the PAOG submodule can adaptively learn sampling points based on the distribution of feature maps at different resolutions.

*4.6.3 Analysis of In-the-wild Videos.* We selected six frames from two videos to compare the predicted 3D human poses by two lifting models, MotionBERT [51] and MotionAGFormer-B [27], with those predicted by our proposed model. The differences have been circled in orange.

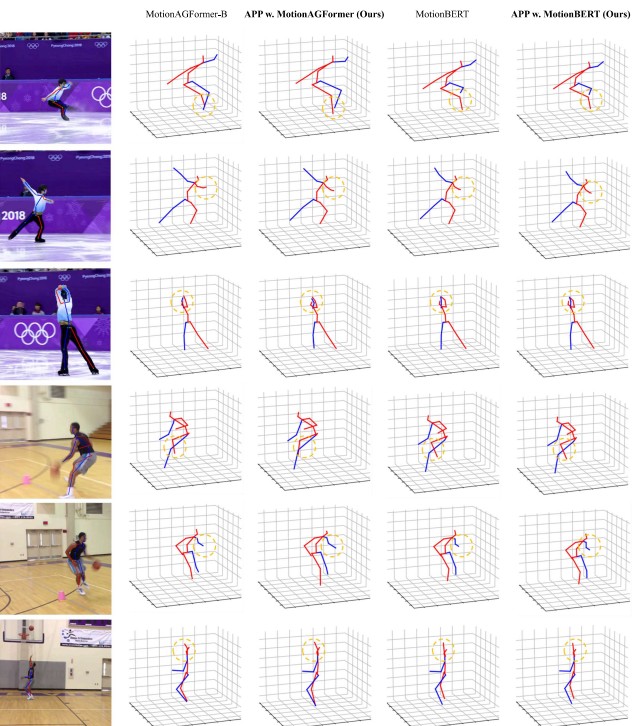

**Figure 5: Qualitative comparison with SOTA methods on in-the-wild images. We select MotionBERT[51] and MotionAGFormer-B [27] as foundation models.**

## 5 CONCLUSION

Presenting APP, short for Adaptive Pose Pooling, a flexible module crafted to mesh with various multi-frame lifting models. This versatile module utilizes the hidden features extracted by lifting models with the feature maps obtained from the 2D pose detector. APP excels in simultaneously extracting spatial and temporal features through this collaborative approach module, and our visualization demonstrates this. Extensive experimentation highlights the reliability of our module, showcasing its ability to enhance the performance metrics of lifting models while consistently achieving state-of-the-art results across diverse scenarios. Notably, APP module presents remarkable adaptability, maintaining its effectiveness even when paired with different 2D poses. A noteworthy feature is its capacity to enhance performance without requiring structural changes to the existing lifting model architecture.

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
