# OpenReview forum: "APP: Adaptive Pose Pooling for 3D Human Pose Estimation from Videos"
_acmmm.org/ACMMM/2024/Conference — MM2024 Poster_

### Official Review · Reviewer_m2Tb · 2024-05-15

**Rating:** 4
**Confidence:** 4

**Summary:**

This paper presents the Adaptive Pose Pooling (APP) module, designed to enhance 3D human pose estimation in videos by integrating latent features from both 2D pose detector and 2D pose lifting model within a multi-frame framework. The efficacy of the APP module is substantiated through comprehensive experiments on the Human3.6M and MPI-INF-3DHP datasets, demonstrating notable accuracy improvements compared to existing methods.

**Strengths:**

1. The simple motivation and straightforward approach make the paper easy to follow.
2. The proposed approach outperforms the state-of-the-art method on benchmark datasets.
3. Some nice visualizations in the experimental sections.

**Limitations:**

1. The paper does not adequately explain why incorporating image features enhances model performance. It appears that these features might be used to refine the estimates from the 2D pose detector for more stable 3D pose predictions.
2. It would be beneficial to explore the impact of using latent features from APP as both the query and value in the Multi-Head Cross Attention (MHCA) setup. What outcomes would this alteration yield?
3. Table 2 should specify the 2D pose detector used by each method for clarity and fair comparison. Most referenced methods utilize CPN or SH detectors, whereas this study employs HRN. Notably, the APP and MotionAGFormer-B demonstrate similar results on the Human3.6M dataset under the same conditions.
4. In Table 6, it is unclear what serves as the query for MHCA in the first configuration. Does it directly employ the hidden features from the 2D pose detector as the key? Additionally, comparing the fifth and last items shows that the influence of APP may not be as significant as expected. To improve readability and facilitate better comparison, assigning serial numbers to each experimental configuration in the table is recommended.

**Suitability:**

3

---

### Official Review · Reviewer_yEae · 2024-05-22

**Rating:** 4
**Confidence:** 3

**Summary:**

The paper presents a flexible plug-and-play module, adaptive pose pooling, for human pose estimation from videos. The method proposes a pose-aware offsets generation module and a pose-aware sampling module, enhancing the model with features from the previous stage of lifting. With the help of a spatio-temporal information fusion module, the method efficiently captures both temporal and spatial features. Experimental results demonstrate the superiority of the method in the HPE task, achieving state-of-the-art performance on two datasets when combined with MotionBERT and MotionAGFormer-B.

**Strengths:**

1. Sophisticated method design: The method embraces a pose-aware sampling module and a pose-aware offsets generation module to learn both temporal and spatial features of human pose from videos. It is well-designed and elegant.
2. Effective experimental results: The experimental results show the effectiveness of the method. The paper provides a detailed comparison with state-of-the-art methods on two datasets, demonstrating superior performance on Human3.6M and MPI-INF-3DHP.
3. Great generalization of the plug-and-play module: The plug-and-play module has excellent generalization capabilities, potentially bringing strong features for future work in the field of human pose estimation.

**Limitations:**

1. Qualitative visualization: In Figure 5, the visualizations for MotionBERT and APP with MotionBERT are not clear. They look the same and are hard to distinguish. Additionally, a column showing the ground truth visualization should be included.
2. Experimental design: Currently, the results mainly include the baseline method. However, the focus should be on how the plug-and-play module boosts the performance of the baseline method. Table 4 only contains a subset of baselines.
3. Paper presentation: The tables currently occupy too much space, making the paper hard to read. To address this, Table 1 can be merged with Table 4 into a single table. The results for different categories can be placed in the supplementary materials. The results on different metrics should be presented in a single-column table, which would be better for demonstration.

**Suitability:**

3

---

### Official Review · Reviewer_xBi9 · 2024-05-25

**Rating:** 3
**Confidence:** 2

**Summary:**

his paper proposes Adaptive Pose Pooling model to estimate 3D human poses from videos.
The model uses 2D features and multi-level feature maps as inputs to obtain spatial and temporal information. Experiments on two datasets demonstrate the effectiveness of the proposed model.

**Strengths:**

The extensive qualitative comparisons on the Human3.6M dataset demonstrate the effectiveness of the proposed model.

**Limitations:**

Why in Table 2 the performance of MotionBERT is better than the proposed model?

**Suitability:**

2

---

### Meta-Review · Area_Chair_pLtA · 2024-07-03

**Recommendation:** Accept (Poster)
**Confidence:** 5

**Metareview:**

The paper present an original Adaptive Pose Pooling model to estimate 3D human poses from videos, which is clearly presented and extensively validated.  The plug-and-play module has excellent generalization capabilities and represent a solid contribution to the community.
The rebuttal addressed the reviewers concerns and all reviewers agreed on accepting this paper.